# microRNA in Extracellular Vesicles Released by Damaged Podocytes Promote Apoptosis of Renal Tubular Epithelial Cells

**DOI:** 10.3390/cells9061409

**Published:** 2020-06-05

**Authors:** Jin Seok Jeon, Eunbit Kim, Yun-Ui Bae, Won Mi Yang, Haekyung Lee, Hyoungnae Kim, Hyunjin Noh, Dong Cheol Han, Seongho Ryu, Soon Hyo Kwon

**Affiliations:** 1Division of Nephrology, Soonchunhyang University Seoul Hospital, Seoul 04401, Korea; jeonjs@schmc.ac.kr (J.S.J.); deepseahk@schmc.ac.kr (H.L.); hkim@schmc.ac.kr (H.K.); nohneph@schmc.ac.kr (H.N.); handc@schmc.ac.kr (D.C.H.); 2Hyonam Kidney Laboratory, Soonchunhyang University Seoul Hospital, Seoul 04401, Korea; wonmi0314@naver.com; 3Soonchunhyang Institute of Med-bio Science (SIMS), Soonchunhyang University, Cheonan, Chungchung nam do 31151, Korea; rainybell@naver.com (E.K.); baeyunui@gmail.com (Y.-U.B.); 4Department of Physiology, Keimyung University School of Medicine, Daegu, Kyungsang buk do 42601, Korea

**Keywords:** extracellular vesicles, microRNA, podocyte, tubule

## Abstract

Tubular injury and fibrosis are associated with progressive kidney dysfunction in advanced glomerular disease. Glomerulotubular crosstalk is thought to contribute to tubular injury. microRNAs (miRNAs) in extracellular vesicles (EVs) can modulate distant cells. We hypothesized that miRNAs in EVs derived from injured podocytes lead to tubular epithelial cell damage. As proof of this concept, tubular epithelial (HK2) cells were cultured with exosomes from puromycin-treated or healthy human podocytes, and damage was assessed. Sequencing analysis revealed the miRNA repertoire of podocyte EVs. RNA sequencing identified 63 upregulated miRNAs in EVs from puromycin-treated podocytes. Among them, five miRNAs (miR-149, -424, -542, -582, and -874) were selected as candidates for inducing tubular apoptosis according to a literature-based search. To validate the effect of the miRNAs, HK2 cells were treated with miRNA mimics. EVs from injured podocytes induced apoptosis and p38 phosphorylation of HK2 cells. The miRNA-424 and 149 mimics led to apoptosis of HK2 cells. These results show that miRNAs in EVs from injured podocytes lead to damage to tubular epithelial cells, which may contribute to the development of tubular injury in glomerular disease.

## 1. Introduction

Glomerular podocytes are terminally differentiated cells and form the filtration barrier between endothelial cells and the basement membrane [1]. Stress conditions, such as high glucose, lead to podocyte injury, which is the first step in diabetic kidney disease, minimal change disease (MCD), and focal segmental glomerular sclerosis (FSGS) [2]. Injured podocytes are associated with glomerulosclerosis, which is the major pathologic process for progression to chronic kidney disease in humans [3,4,5]. Glomerular injury leads to tubulointerstitial inflammation and fibrosis, which is a common final pathway in kidney disease [6,7,8]. However, the mechanism of tubulointerstitial injury in glomerular disease remains unclear.

Extracellular vesicles (EVs) refer to all membrane-bound vesicles released from cells into the extracellular space [9,10]. Cells in the nephron constitutively release EVs under healthy conditions [11]. Studies of the biological role of EVs in intercellular communication in the nephron have indicated that they can act as messengers [2,12,13,14,15]. EVs from parent cells specifically interact with recipient cells in the nephron [16], and recent studies suggested a proximal to distal signaling pathway [12,14,17,18,19].

MCD and FSGS are primary podocytopathies, and the underlying abnormality is extensive podocyte foot process effacement as observed by electron microscopy [20]. However, the prognosis and treatment for FSGS and MCD are quite different. Most nephrotic syndromes due to MCD respond to corticosteroid therapy. In contrast, patients with FSGS have higher rates of corticosteroid resistance and progression to chronic kidney disease leading to end-stage renal disease [21]. A recent study revealed different urinary EV microRNAs (miRNAs) in FSGS compared to in MCD [22]. miR-193a in urinary EVs was suggested as a biomarker for FSGS [23]. FSGS-specific miRNAs can act as biomarkers and may play a role in renal injury. miRNAs regulate gene expression via post-transcriptional mechanisms. Furthermore, miRNAs in EVs may play a role in the pathology of various diseases [24]. miRNAs are frequently dysregulated in the development of renal fibrosis [25]. Further, EV-enclosed miRNAs are potential biomarkers of renal fibrosis [26]. miRNAs in podocyte EVs of patients with FSGS may contribute to tubulointerstitial fibrosis and lead to renal failure.

We hypothesized that miRNAs in EVs from injured podocytes could lead to damage of tubular epithelial cells in the nephron. Therefore, we induced podocyte damage using puromycin aminonucleoside (PAN) and co-incubated proximal epithelial cells with EVs isolated from podocytes. miRNAs in EVs from podocytes were analyzed to investigate the function of podocyte injury-related miRNAs.

## 2. Materials and Methods

### 2.1. Human Podocyte Culture

Conditionally immortalized human (i.e., wild-type) podocytes, a gift from Dr. Moin Saleem (University of Bristol, Bristol, UK), were cultured as described previously [27,28]. Briefly, the cells were grown on culture plates in RPMI 1640 medium supplemented with EV-free 10% fetal bovine serum (Invitrogen, Carlsbad, CA, USA) and penicillin–streptomycin solution (1:100, Invitrogen). Podocytes were propagated at 33 °C in the presence of 10 U/mL recombinant human γ-interferon (Invitrogen). We did not induce podocyte differentiation. To induce podocyte injury, the cells were grown in a 100-mm culture dish and treated with 25 µg/mL PAN (Sigma-Aldrich, St. Louis, MO, USA) for 48 h.

### 2.2. EV Isolation and Size Distribution

EVs released from podocytes were isolated using a commercial kit (ExoQuick-TC™; System Biosciences, Palo Alto, CA, USA) according to the manufacturer’s guidelines. The culture medium (10 mL) was centrifuged at 3000× *g* for 15 min to remove the cells and cell debris. The supernatants were mixed with 2 mL ExoQuick-TC reagent and incubated overnight at 4 °C. After incubation, the samples were centrifuged at 1500× *g* for 30 min and the supernatants were aspirated. The pellets containing EVs were resuspended in 100–200 μL of sterile phosphate-buffered saline (PBS).

The size of the EVs was determined by nanoparticle tracking analysis using a Nanosight NS300 (Malvern Instruments Ltd., Malvern, UK) in size mode with a 488-nm blue laser module and sCMOS camera. Samples were diluted in particle-free PBS (0.2-µm filtered) to a final volume of 1 mL. The following settings were used according to the manufacturer’s instructions for nanoparticle tracking analysis using version 3.4 Build 3.4.003 with standard measurements; the level of the camera was 15, the number of gain was 366, and the temperature was 25 °C. The exposure time was automatically set in the program. Further settings, such as viscosity to water of approximately 0.80–0.90 cP, minimum track length, and minimum expected size, were automatically set.

### 2.3. Proximal Tubule Cell Culture and EV Treatment

The human proximal tubule HK2 epithelial cell line was purchased from the American Type Culture Collection (Manassas, VA, USA) and cultured at 37 °C in a 5% CO_2_ atmosphere in Dulbecco’s modified Eagle’s medium mixed 1:1 (*v/v*) with F12 medium supplemented with 10% EV-free fetal bovine serum (all from Life Technologies, Carlsbad, CA, USA). Near-confluent cells were incubated in serum-free medium for 24 h to arrest the cells and synchronize the cell cycle. HK2 cells were co-cultured with EVs from the podocyte culture (with or without PAN) for 24 h.

### 2.4. EV Internalization Assay

To assess EVs internalization, HK2 cells were incubated with fluorescently labelled EVs and analyzed by confocal microscopy (Olympus, Tokyo, Japan). Podocyte EVs were labelled with the red fluorescent dye PKH26 (Sigma-Aldrich) for 5 min at room temperature according to the manufacturer’s instructions. Labelled EVs were then washed twice by centrifugation (20,000× *g* 20 min at 4 °C) and re-suspended in PBS. HK2 cells were seeded onto glass coverslips and treated with EVs (10 µg/mL) for 3 h at 37 °C. HK2 cells were washed three times with cold PBS, fixed for 10 min in 4% paraformaldehyde with 0.3% Triton X-100, and washed three times in PBS. The fixed cells were incubated with Alexa Fluor 488 phalloidin (1:200, Thermo Fisher Scientific, Waltham, MA, USA; A12379). Nuclei were stained with 4′,6-diamidino-2-phenylindole (DAPI) using ProLong Gold Antifade Mountant (Thermo Fisher Scientific; P36935). Images were captured using a fluorescence microscope (Olympus).

### 2.5. Western Blotting

EVs and HK2 cells were subjected to Western blot analyses using standard procedures. The membranes were immunoblotted with antibodies against the tumor susceptibility gene 101 (1:2000, Abcam, Cambridge, UK), ALIX (1:1000, Cell Signaling Technology, Danvers, MA, USA), cleaved poly (ADP-ribose), polymerase (1:1000, Cell Signaling Technology), caspase-3 (1:1000, Cell Signaling Technology), phosphorylated extracellular signal-regulated kinase (pERK) (1:1000, Cell Signaling Technology), total (t)ERK (1:1000, Cell Signaling Technology), p-p38 (1:1000, Cell Signaling Technology), tp38 (1:000, Cell Signaling Technology), E-cadherin (1:1000, BD Biosciences, Franklin Lakes, NJ, USA), fibronectin (1:2000, Abcam), collagen IV (1:1000, Southern Biotech, Birmingham, AL, USA), α-smooth muscle actin (1:1000, Abcam), and β-actin (1:5000, Sigma-Aldrich). Following incubation with the primary antibodies, the membranes were washed in TBS-T and incubated with horseradish peroxidase-conjugated anti-rabbit or anti-goat (collagen IV) secondary antibodies.

### 2.6. Flow Cytometry

HK2 cells treated with EVs were stained for 20 min with Annexin V (BD Biosciences) followed by incubation with a fluorescein isothiocyanate- or phycoerythrin-conjugated secondary antibody. Apoptosis was assessed using a FlowSight (Luminex, Austin, TX, USA) flow cytometer.

HK2 cells were seeded into 6-well plates at 1 × 10^6^ cells per well. After transfection and incubation for 2 days, the cells were harvested. Apoptosis was evaluated using an Annexin V apoptosis detection kit (eBioscience, San Diego, CA, USA) according to the manufacturer’s instructions. The cells were washed once with 100 µL binding buffer and stained for 10 min with Annexin V at room temperature in the dark. Stained cells were washed once with 200 µL binding buffer and stained again with 7-aminoactinomycin (7-AAD) D or propidium iodide to evaluate cell viability. Stained cells were analyzed using a FACSCanto II (BD Biosciences).

### 2.7. miRNAs Extraction From EVs

RNA was extracted from the EVs using a miRNeasy Mini Kit (Qiagen, Hilden, Germany). EV suspensions (200 μL) were mixed with QIAzol lysis buffer (1 mL), and the mixtures were processed according to the manufacturer’s guidelines. RNA was eluted in RNase-free water (20 μL). Purified RNA was analyzed using an Agilent 2100 Bioanalyzer with RNA Pico and Small RNA kits to examine the size distribution of the RNAs of EVs (Agilent Technologies, Santa Clara, CA, USA).

### 2.8. cDNA Library Preparation and Small RNA Sequencing

The samples were processed to produce EVs RNA (10 ng) as input for each library. Small RNA libraries were constructed using a SMARTer smRNA-Seq Kit for Illumina^®^ (Takara Bio, Shiga, Japan) according to the manufacturer’s guidelines. We generated sequencing libraries by polyadenylation, cDNA synthesis, and polymerase chain reaction amplification. Equimolar amounts of the libraries were pooled and sequenced on an Illumina^®^ HiSeq 2500 instrument (Illumina, San Diego, CA, USA) to generate 101 base reads. Next-generation sequencing analysis was performed by Macrogen (Seoul, Korea).

### 2.9. Analysis of RNA Sequencing Data

We performed sequence alignment and detected known and novel miRNAs using the miRDeep2 software algorithm. Prior to aligning the sequences, we retrieved *Homo sapiens* reference genome release hg19 from the UCSC Genome Browser, which we indexed using Bowtie (1.1.2), a program for aligning experimental and reference sequences. The reads were then aligned to mature and precursor miRNAs from *H. sapiens* obtained from miRBase 21.

### 2.10. miRNA Mimic Treatments

HK2 cells were seeded at 1 × 10^6^ cells/well in 6-well culture plates. After incubation for 1 day, the cells were transfected with 40 nM of negative control or mimic using Lipofectamine RNAiMax (Invitrogen) in Opti-MEM (Invitrogen) according to the manufacturer’s protocol. After 2 h, the media were replaced with Dulbecco’s modified Eagle’s medium and incubated for the indicated time periods.

### 2.11. Statistical Analyses

Data are presented as the means ± standard error of the mean. Statistical analyses were conducted using SPSS version 22.0 (SPSS, Inc., Chicago, IL, USA). Parameters were evaluated by one-way analysis of variance (ANOVA) with Tukey’s post-hoc test. Values of *p* < 0.05 were considered as statistically significant.

## 3. Results

### 3.1. Characterization of Podocyte EVs

Podocyte EVs were characterized by nanoparticle tracking analysis. The podocyte EVs were 101.2 ± 51.1 nm in size and were homogeneous (Figure 1A, Appendix A).

### 3.2. Podocytes Release EVs

PAN decreased the number and viability of podocytes (Figure 1B,C). PAN induced the apoptosis of podocytes and decreased cells viability (Figure 1D,E). After PAN treatment, podocytes released more EVs into the supernatant (Figure 1F).

### 3.3. HK2 Cells Interact With EVs From Podocytes

To determine whether podocyte EVs played a role in glomerular tubular crosstalk, we assessed EV internalization by HK2 cells. In the confocal microscopy images, labelled EVs were visible on the surface of HK2 cells. This suggests a physical interaction between EVs from podocytes and HK2 cells (Figure 2).

### 3.4. EVs from Injured Podocytes Induce Apoptosis of HK2 Cells

EVs from PAN-treated podocytes decreased the number of HK2 cells after co-incubation (Figure 3A). Cleaved poly (ADP-ribose) polymerase (PARP) was increased in HK2 cells by EVs from PAN-treated podocytes (Figure 3B,C). Healthy podocyte derived EVs did not change the expression of cleaved PARP in HK2 cells. This suggests that injured podocyte-derived EVs induced the apoptosis of HK2 cells. Using flow cytometry and Western blotting, we assessed the changes of HK2 cells after co-incubation with EVs. Flow cytometry showed an increase in the number of apoptotic HK2 cells treated with EVs from PAN-treated podocytes (Figure 3D,E).

### 3.5. EVs from Injured Podocytes Stimulate the ERK and p38 Pathways in HK2 Cells

Podocyte-derived EVs also increased the expression of fibronectin and collagen IV in HK2 cells (Figure 4A–E). Additionally, non-injured podocyte EVs did not change the expression of fibronectin and collagen IV of HK2 cells. Activation of the ERK and p38 signaling pathways is associated with tubulointerstitial fibrosis and apoptosis in kidney disease [29,30,31]. Therefore, we explored the effect of podocyte EVs on this pathway. Phosphorylation of p38 was robust in HK2 cells exposed to PAN-treated podocyte EVs (Figure 4A,D). The pattern of ERK activation was similar to that of the p38 pathway (Figure 4D,E). PAN-treated podocyte EVs also increased the expression of fibronectin and collagen IV in HK2 cells (Figure 4B,C). Taken together, only PAN-treated podocyte-derived EVs induced apoptosis, which was associated with the p38 and ERK pathways.

### 3.6. EV miRNA Profile from Injured Podocytes Differs from That of Non-Injured Podocytes

miRNAs are associated with renal fibrosis [25]. EVs are also transferred from the proximal tubular cells to the distal tubule and collect duct cells [12]. Thus, we hypothesized that podocyte-derived EVs miRNAs played a role in glomerular tubular crosstalk. miRNAs profiles in EVs from podocytes, with or without PAN treatment, were assessed by RNA sequencing. The heatmap showed different miRNA expression patterns in podocyte EVs with or without PAN treatment (Figure 5A). RNA sequencing revealed 134 differentially expressed miRNAs between PAN-treated and non-treated podocyte-released EVs. Among the 134 identified miRNAs, 63 from the PAN-treated group were upregulated and 71 were downregulated compared to those in the untreated group. Figure 5B shows the top 20 miRNAs.

### 3.7. miR-424 and 149 Induces HK2 Cell Apoptosis

Among the upregulated miRNAs, we focused on miR-149-5p, 424-5p, 542-3p, 582-5p, and 874-3p. These miRNAs are associated with apoptosis and inhibit cell proliferation [32,33,34,35,36]. Among them, mimics of miR-424-5p and 149-5p induced apoptosis of HK2 cells (Figure 5C,D) according to flow cytometry. We evaluated the phenotype change and apoptosis pathway of HK2 cells after treatment with miR-424-5p and 149-5p mimics. Both mimics increased cleaved PARP expression (Figure 5F) and fibronectin expression (Figure 5G). Collagen expression in HK2 cells did not increase after treatment with both mimics (Figure 5H). Only miR-424-5p was associated with the p38 pathway (Figure 5I). miR-424-5p and 149-5p did not alter the ERK pathway (Figure 5J). These data suggest that miRNAs in injured podocyte-derived EVs partially contribute to the effect of tubular damage between glomerular tubular crosstalk.

## 4. Discussion

Traditionally, proteinuria from glomerular damage is considered to cause tubulointerstitial injury and lead to kidney disease progression [37]. In the current study, we found that EVs from injured podocytes induced tubular epithelial cell damage to the pathway that was activated in these cells. In addition, miRNAs in EVs may have provoked this change. Taken together, our findings indicate that EVs mediating glomerular–tubular crosstalk and miRNAs in EVs partially contribute to this change. Podocytes under stress conditions may alter the glomerulotubular connection and have clinical relevance in human glomerular disease.

A previous study suggested that microparticles from podocytes induced tubulointerstitial fibrosis [13], which is consistent with our report. However, other EVs such as exosomes from podocytes did not affect the tubulointerstitium (tubule fibrotic signaling) in their study. Rather, they showed that microparticles from untreated podocytes (without injury) caused changes in tubular cells. This differs from the present study in which EVs from healthy podocytes did not affect tubular epithelial cells (HK2 cells). We first induced podocyte injury and then confirmed that EVs from these cells expressed tumor susceptibility gene 101. This supports the theory that injured but not healthy podocytes induce tubulointerstitial damage and contribute to the pathogenesis of disorders such as FSGS and diabetic kidney disease.

EVs from injured podocytes showed different characteristics compared to non-stressed podocytes. First, the number of EVs was elevated after injury. Several other studies also demonstrated that stressed cells release more EVs [38,39,40]. Second, EVs from injured podocytes had unique miRNA profiles compared to non-injured podocyte EVs. This suggests that stress conditions change not only the number but also the content of EVs. Tubular cells under hypoxia released EVs carrying transforming growth factor-β-1 mRNA [40]. These EVs activated fibroblasts in the nephron of a unilateral ureter obstruction model. This model also suggests that EV-mediated injury in the nephron is efficient in the early state of injury. In this study, we did not identify different mRNAs or proteins in EVs under stress conditions. The different expression of miRNAs in EVs was also suggested as a biomarker [24]. We previously identified differences in miRNAs in EVs between control and hypertensive [41], obese [42], or diabetic nephropathy [43] patients. These differences under stress conditions may have functional implications in terms of disease progression or complications.

We focused on specific miRNAs to determine whether they mimicked the effect of EVs of injured podocytes. In our study, miR-424-5p and 149-5p mimics induced apoptosis of tubular cells. Thus, these miRNAs may participate in glomerulotubular crosstalk in EVs. Using tumor cells, miR-424-5p was suggested to suppress cell proliferation, migration, and tube formation capabilities [44]. miR-149-5p also induced cell apoptosis in the pancreas [45]. Interestingly, both miR-424 and miR-149 suppressed the proliferation and survival of renal cancer cells [46,47]. This suggests that these microRNAs have unique functions in the nephron. In addition, miR-424 is decreased in the kidney after tubule injury and reduced in patients with type 1 diabetes and microalbuminuria compared to in those without albuminuria [48,49].

We identified a total of 134 differentially expressed miRNAs in injured podocytes. Among them, miR-200b was upregulated in EVs released from injured podocytes. Similarly, miR-200c was higher in urine from patients with minimal change disease and focal segmental glomerulosclerosis compared to in healthy kidney donors [50]. Urinary miR-200b and miR-200c were correlated with renal function in minimal change disease or focal segmental sclerosis [50]. This suggests that the miR-200 family has biological functions in nephrotic syndrome.

Mitogen-activated protein kinases, including ERK1/2 and p38, coordinate responses that dictate cell death or survival [51]. Activation of the p38 pathway is associated with renal fibrosis, whereas inhibition attenuated renal atrophy and fibrosis in a murine model of renal artery stenosis [52]. In our experiments, podocyte-derived EVs activated the ERK and p38 pathways. However, no dramatic change was observed in the ERK pathway after treatment with miR-424-5p and 149-5p mimics. This indicates that miRNAs in EVs partially contribute to this crosstalk. Other materials, such as other RNAs, proteins, and metabolites, may coordinate with miRNAs. However, we did not evaluate all microRNAs of injured podocyte EVs. Therefore, further studies of candidate microRNAs in exosomes are necessary to evaluate glomerulotubular crosstalk.

Our study had some limitations. First, we did not confirm whether microRNAs in podocyte-derived EVs were transferred to HK2 cells. Second, other materials such as mRNA and protein in podocyte-derived EVs may have induced HK2 cells. This study focused on the miRNAs of injured podocyte-derived EVs compared to those of healthy EVs. Finally, this was an in vitro study to evaluate “glomerulotubular crosstalk”, which should be validated in vivo.

In summary, we demonstrated that EVs from injured podocytes induce tubulointerstitial (tubular cell) apoptosis. This response was dependent upon activation of the p38 pathway. Tubular cell changes, such as apoptosis, may be elicited by miRNAs in EVs from podocytes. This pathway of glomerulotubular crosstalk may explain the decline in renal function in glomerular disease.

## Figures and Tables

**Figure 1 cells-09-01409-f001:**
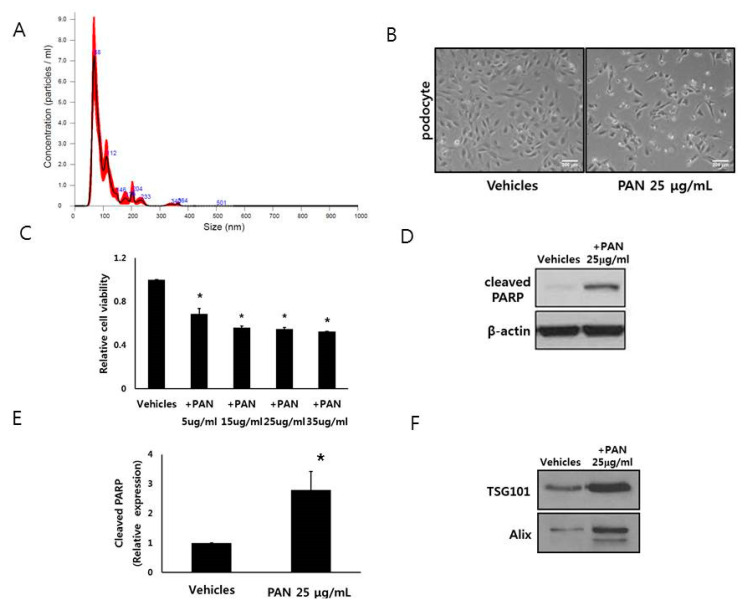
Podocytes release extracellular vesicles (EVs). (**A**) Size distribution of isolated podocyte EVs as determined by nanoparticle tracking analysis. (**B**,**C**) Number of podocytes after PAN exposure was decreased. (**D**,**E**) PAN treatment (25 μg/mL) induced apoptosis of podocytes. (**F**) Number of EVs from PAN-exposed podocytes was more than that of control podocytes. HP; human podocyte, PAN; puromycin aminonuceloside. * *p* < 0.05 vs. HP control.

**Figure 2 cells-09-01409-f002:**
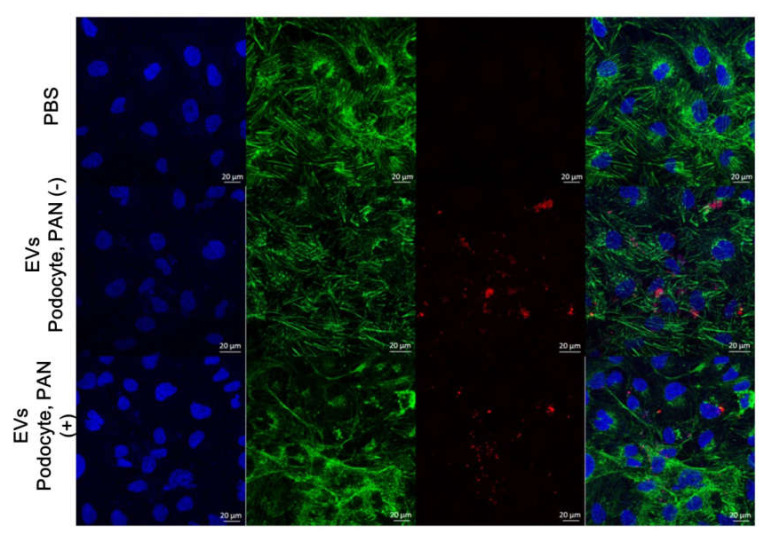
HK2 cells interact with EVs. Confocal microscopy showed that EVs of PKH26-labelled (red) from podocytes (PAN-treated or untreated) were visible with HK2 cells. Filamentous actin was stained with phalloidin (green) and nucleoli were labelled with 4′,6-diamidino-2-phenylindole (DAPI) (blue).

**Figure 3 cells-09-01409-f003:**
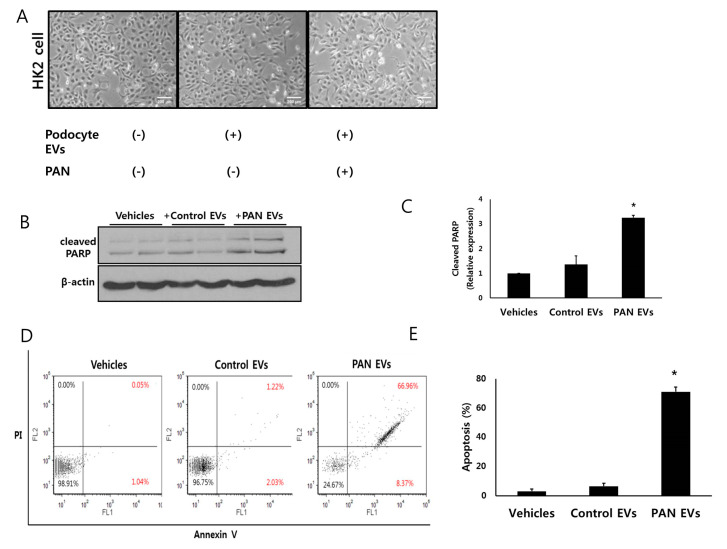
EVs from injured podocytes led to HK2 cell apoptosis (**A**) EVs from PAN-treated podocytes decreased the number of HK2 cells. (**B**,**C**) PAN-treated podocytes released EVs increased cleaved poly (ADP-ribose) polymerase (PARP) expression of HK2 cells **(D,E)** PAN-treated podocytes released EVs induced apoptosis of HK2 cells after co-incubation. PAN; puromycin nucleoside, PI; propidium iodide. * *p* < 0.05 vs. vehicles.

**Figure 4 cells-09-01409-f004:**
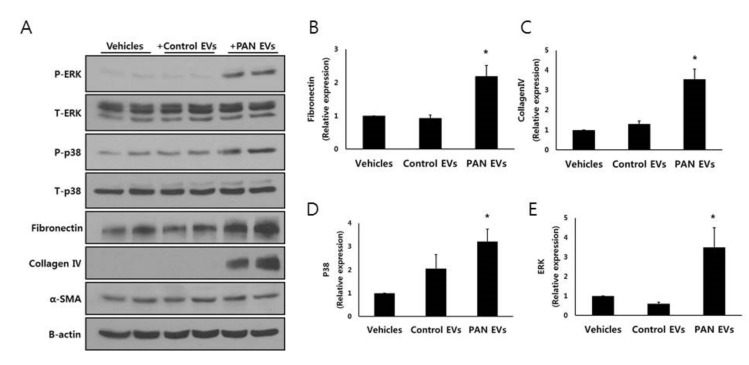
EVs from injured podocytes change HK2 cells. (**A**–**C**) Co-incubation with EVs from PAN-treated podocytes increased expression of apoptosis marker (cleaved poly (ADP-ribose) polymerase (PARP), fibronectin, and collagen IV) of HK2 cells. (**D**,**E**) EVs from PAN-treated podocytes activated the extracellular signal-regulated kinase (ERK) and p38 pathways in HK2 cells. * *p* < 0.05 vs. vehicles.

**Figure 5 cells-09-01409-f005:**
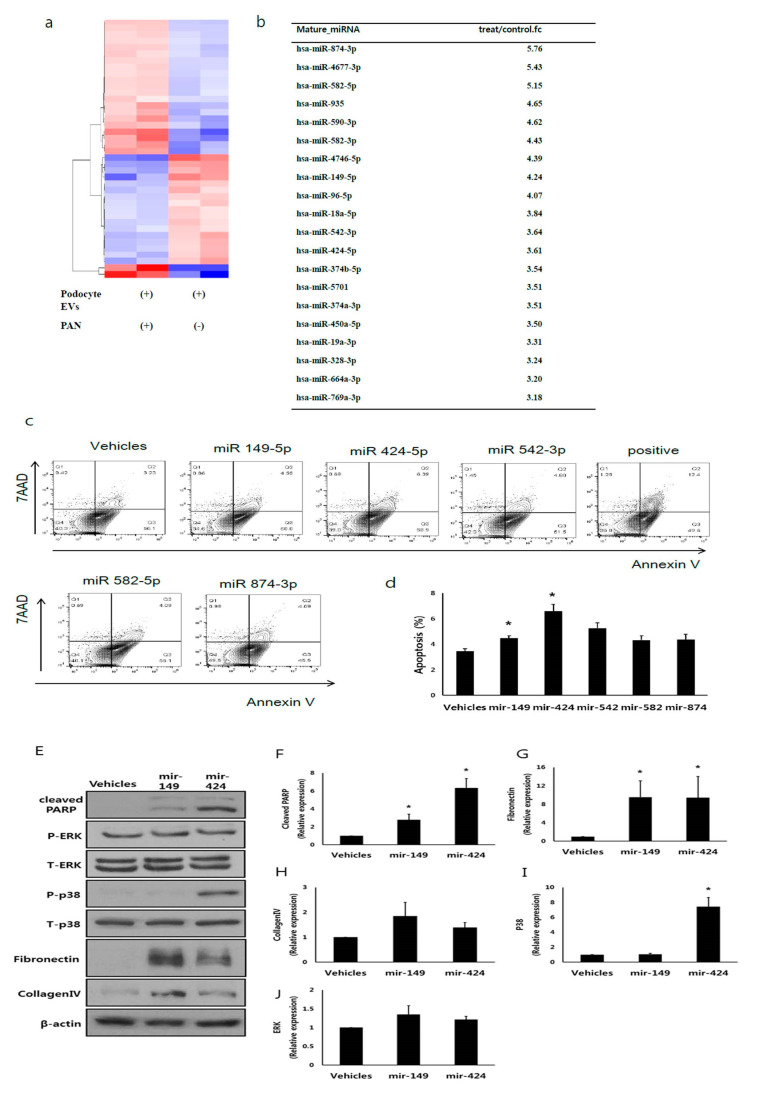
miR-424-5p and miR-149-5p mimics induced apoptosis of HK2 cells. (**A**) Heatmap of podocyte EV miRNA sequencing. (**B**) Top 20 miRNAs among 134 differently identified miRNAs between EVs from healthy podocytes and injured podocytes. (**C**,**D**) Flow cytometry showing the apoptosis of HK2 cells after miRNA-424-5p and miRNA-149-5p mimic treatment. (**E**–**J**) miR-424 mimics lead to HK2 cell apoptosis via p38 pathway. AAD; 7-aminoactinomycin D. * *p* < 0.05 vs. vehicles.

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
