# Peer review of "microRNA in Extracellular Vesicles Released by Damaged Podocytes Promote Apoptosis of Renal Tubular Epithelial Cells"

_cells, 2020, doi:10.3390/cells9061409_

Round 1
Reviewer 1 Report
This is an original manuscript that examines the effects of extracellular vesicles from podocytes on kidney tubule cells. The primary finding is that EVs from damaged podocytes promote apoptosis in renal tubular epithelial cells. The authors further identify miR-424 and 149 as candidate miRNA responsible for the effects of podocyte EVs. The study is novel and well-written and appears methodologically sound. Minor comments are added before. I add that the authors refer to supplementary figures that I was unable to access. As such I was unable to evaluate these figures
- What was the dose of EVs for experiments and how did the authors determine it?
- One question I have is whether the EVs are truly podocyte-derived. No detail was provided with respect to the protocol for inducing differentiation and no evidence is shown to confirm a podocyte-like phenotype in either the cells or the EVs (i.e. expression of nephrin, synaptopodin, podocalyxin…).
- The authors indicate in the methodology that the Nanosight system was employed with a syringe pump. However, no detail is provided with respect to flow rate. Can the authors clarify whether EVs were analyzed under flow conditions or were they analyzed in a static environment after injection. Similarly, what was the exposure time for the camera?
- Please provide concentrations of antibodies used for Western Blot analysis. Also what was the source of the secondary antibodies?
- The authors should clarify the protocol for PKH26 labeling. In the methodology it states that fixed cells were incubated with antibodies to PKH26. I am not aware of any antibody that reacts with PKH26 which is a red lipid membrane dye and not an endogenous protein. Even if such an antibody were to exist, the authors provide no protocol for labeling of EVs with PKH26. Traditionally, EV labeling with PKH26 involves incubation with the dye for a period of time followed by a wash. Imaging of the EVs requires no additional steps to label since the PKH26 itself is fluorescent. Please clarify if an atypical labeling protocol for EVs was used? If so please provide thorough detail on how this was done and the rationale for using this approach.
- What was the justification for using Annexin V/PI to detect apoptosis in one instance (Fig 2) but Annexin V/7-AAD in a separate instance (fig 4)
- The authors state that “injured podocytes release more EVs”. This is certainly possible and consistent with other reports. However I note that the only evidence provided to support this was un-normalized protein data (Figure 1f). Why did the authors not confirm this with NTA-based quantitation or some other means? If the protein data is the only evidence to support this conclusion then I think that the authors may need to soften this statement.
- No details are provided with respect to the confocal microscopy. I note also that a single image is shown. Were the authors able to confirm internalization of EVs with confocal microscopy? Is it not possible that the effects of the EVs were achieved by surface interactions and the miRNA represent an overlapping but independent pathway?
Minor comments:
- There is a small typo on p.2 line 51 “miRNAs regulated gene expression…”. I believe that this should be “miRNAs regulate gene expression”
Author Response
Response list of reviewer’s comments
Reviewer 1
This is an original manuscript that examines the effects of extracellular vesicles from podocytes on kidney tubule cells. The primary finding is that EVs from damaged podocytes promote apoptosis in renal tubular epithelial cells. The authors further identify miR-424 and 149 as candidate miRNA responsible for the effects of podocyte EVs. The study is novel and well-written and appears methodologically sound. Minor comments are added before. I add that the authors refer to supplementary figures that I was unable to access. As such I was unable to evaluate these figures
Thank you
What was the dose of EVs for experiments and how did the authors determine it?
Response: Thank you for your comments, which helped improve our manuscript.
We used EVs from human podocytes grown in a 100-mm dish.
“HK2 cells were seeded at a density of 2 × 104/cm2 and grown in 6-well culture plates. The HK2 cells were treated with EVs from podocytes cultured in a 100-mm dish (80% confluent).“
We have clarified this point in the Methods section.
One question I have is whether the EVs are truly podocyte-derived. No detail was provided with respect to the protocol for inducing differentiation and no evidence is shown to confirm a podocyte-like phenotype in either the cells or the EVs (i.e. expression of nephrin, synaptopodin, podocalyxin…).
Response: Thank you for your comments.
We did not induce differentiation of human podocytes obtained from Dr Moin A. Saleem. In the previous study, podocytes expressed nephrin at 33℃ and 37℃ (J Am Soc Nephrol. 2002 Mar;13(3):630-8). We also previously evaluated undifferentiated podocytes (PLoS One. 2017 Sep 7;12(9):e0184575).
We have clarified this point in the Methods section.
The authors indicate in the methodology that the Nanosight system was employed with a syringe pump. However, no detail is provided with respect to flow rate. Can the authors clarify whether EVs were analyzed under flow conditions or were they analyzed in a static environment after injection. Similarly, what was the exposure time for the camera?
Response: Thank you for your comment.
We have checked this method with Malvern Instruments Korea. Nanosight NS300 does not use a syringe pump system. The sample was loaded using a syringe. The exposure time is set automatically. This information has been included in the Methods section.
Please provide concentrations of antibodies used for Western Blot analysis. Also what was the source of the secondary antibodies?
Response: We have described the antibody concentration in the description of the western blotting procedure in the Methods section.
The authors should clarify the protocol for PKH26 labeling. In the methodology it states that fixed cells were incubated with antibodies to PKH26. I am not aware of any antibody that reacts with PKH26 which is a red lipid membrane dye and not an endogenous protein. Even if such an antibody were to exist, the authors provide no protocol for labeling of EVs with PKH26. Traditionally, EV labeling with PKH26 involves incubation with the dye for a period of time followed by a wash. Imaging of the EVs requires no additional steps to label since the PKH26 itself is fluorescent. Please clarify if an atypical labeling protocol for EVs was used? If so please provide thorough detail on how this was done and the rationale for using this approach.
Response: We agree with this critical point and apologize for the unclear description. We have rewritten this part in the Methods section.
Thank you for your comments.
“To assess exosome internalization, HK2 cells were incubated with fluorescently labelled EVs and analysed by confocal microscopy (Olympus, Tokyo, Japan). Podocyte EVS were labelled with red fluorescent dye, and fixed cells were stained with primary antibodies against PKH26 (Sigma-Aldrich) for 5 min at room temperature according to manufacturer’s instructions. Labelled EVs were washed twice by centrifugation (20,000 ×g, 20 min, 4℃) and re-suspend in PBS before treatment. HK2 cells were seeded onto glass coverslips and treated with EVs (10 µg/mL) for 3 h at 37℃. HK2 cells were washed three times with cold PBS, fixed for 10 min in 4% paraformaldehyde with 0.3# Triton X-100, and washed three times in PBS. Fixed cells were incubated with secondary antibodies conjugated to Alexa Fluor 488 phalloidin (1:200, Thermo Fisher Scientific, Waltham, MA, USA; A12379). Nuclei were stained with 4',6-diamidino-2-phenylindole using ProLong Gold Antifade Mountant (Thermo Fisher Scientific; P36935). Images were captured with a fluorescence microscope (Olympus).”
What was the justification for using Annexin V/PI to detect apoptosis in one instance (Fig 2) but Annexin V/7-AAD in a separate instance (fig 4)
Response: Although the advantage of 7-AAD over PI is that there is minimal spectral overlap with most fluorescence probes, both PI and 7-AAD are useful for cell viability analysis by flow cytometry. Two independent laboratories conducted flow cytometry experiments and used different protocols to evaluate cell viability.
The authors state that “injured podocytes release more EVs”. This is certainly possible and consistent with other reports. However I note that the only evidence provided to support this was un-normalized protein data (Figure 1f). Why did the authors not confirm this with NTA-based quantitation or some other means? If the protein data is the only evidence to support this conclusion then I think that the authors may need to soften this statement.
Response: We agree with your comments.
We have changed from “Injured podocytes released more EVs” to “ Podocytes released EVs”
No details are provided with respect to the confocal microscopy. I note also that a single image is shown. Were the authors able to confirm internalization of EVs with confocal microscopy? Is it not possible that the effects of the EVs were achieved by surface interactions and the miRNA represent an overlapping but independent pathway?
Response: The methods used for confocal microscopy have been added to the Methods section. We agreed with your comments. This image suggests that a physical interaction occurs between EVs and HK2 cells. We have included this information in the Results section and clarified this as a limitation.
“Our study had some limitations. First, we did not confirm whether miRNAs in podocyte-derived EVs were transferred to HK2 cells. Second, other materials such as mRNA and proteins in podocyte-derived EVs may have induced HK2 cells. This study focused on the miRNAs of injured podocyte-derived EVs compared those of healthy EVs.”
Minor comments:
There is a small typo on p.2 line 51 “miRNAs regulated gene expression…”. I believe that this should be “miRNAs regulate gene expression”
Response: The phrase has been changed accordingly.
Reviewer 2 Report
Here, Jeon and co-workers investigated the role of miRNAs in EVs that are released by damaged podocytes to promote apoptosis in renal tubular epithelial cells. Although, the topic may interest researchers, this manuscript is far from being considered for publication. Here are my comments:
- The language of this manuscript needs to be thoroughly checked before resubmission.
- I failed to understand a solid rationale behind testing the role of miRNAs in EVs.
- Please provide pro-caspase data for figure 1D.
- The result section is written extremely poorly. Please explain everything in detail before submitting it again.
- Can the authors explain why PAN treated cells take up less EV?
- How did the authors come upon miRs-149 and 424? Where are the mimic assay data?
- Is the data reproducible in other cell lines or in vivo?
Author Response
The language of this manuscript needs to be thoroughly checked before resubmission.
Response: A native English speaker has edited this manuscript again.
I failed to understand a solid rationale behind testing the role of miRNAs in EVs.
Response: Thank you for your comment, which helped to improve our manuscript. We have added our rational for evaluating miRNAs in the Introduction section.
Please provide pro-caspase data for figure 1D.
Response: Your comment is valuable for evaluating the pathway of apoptosis. However, we did not examine the caspase pathway. Because of the limited revision time (10 days), further experiments were not possible. However, cleavage of PARP is thought to be required for later stages of apoptosis (Role of Poly(ADP-ribose) Polymerase (PARP) Cleavage in Apoptosis. Caspase 3-resistant PARP mutant increases rate of apoptosis in transfected cells J Biol Chem. 1999 Aug 13;274(33):22932-40. doi: 10.1074/jbc.274.33.22932.)
The result section is written extremely poorly. Please explain everything in detail before submitting it again.
Response: We have revised the Results section and included additional details.
Can the authors explain why PAN treated cells take up less EV?
Response: PAN-treated podocytes appeared to release more EVs. However, quantification of EVs was not performed using the NTA-based method. We have clarified this point in the revised manuscript.
HK2 cells were not treated with PAN in this study.
How did the authors come upon miRs-149 and 424? Where are the mimic assay data?
Response: MiRNAs in EVs are associated with renal fibrosis. We hypothesized that miRNAs in EVs play a role in communication between podocytes and tubular cells. We analysed miRNAs in EVs of podocyte-derived EVs RNA sequencing identified 134 miRNAs differentially expressed in injured podocyte-derived EVs compared to in non-injured podocyte EVs. Among them, we focused on 5 miRNAs after literature review. These miRNAs are known to be involved role in apoptosis or inhibit cell proliferation.
Uchino, K.; Takeshita, F.; Takahashi, R.U.; Kosaka, N.; Fujiwara, K.; Naruoka, H.; Sonoke, S.; Yano, J.; Sasaki, H.; Nozawa, S., et al. Therapeutic effects of microRNA-582-5p and -3p on the inhibition of bladder cancer progression. Mol Ther 2013, 21, 610-619, doi:10.1038/mt.2012.269.
Wang, K.; Liu, F.; Zhou, L.Y.; Ding, S.L.; Long, B.; Liu, C.Y.; Sun, T.; Fan, Y.Y.; Sun, L.; Li, P.F. miR-874 regulates myocardial necrosis by targeting caspase-8. Cell Death Dis 2013, 4, e709, doi:10.1038/cddis.2013.233.
Bischoff, A.; Huck, B.; Keller, B.; Strotbek, M.; Schmid, S.; Boerries, M.; Busch, H.; Muller, D.; Olayioye, M.A. miR149 functions as a tumor suppressor by controlling breast epithelial cell migration and invasion. Cancer Res 2014, 74, 5256-5265, doi:10.1158/0008-5472.CAN-13-3319.
Yang, L.; Dai, J.; Li, F.; Cheng, H.; Yan, D.; Ruan, Q. The expression and function of miR-424 in infantile skin hemangioma and its mechanism. Sci Rep 2017, 7, 11846, doi:10.1038/s41598-017-10674-7.
Yuan, L.; Yuan, P.; Yuan, H.; Wang, Z.; Run, Z.; Chen, G.; Zhao, P.; Xu, B. miR-542-3p inhibits colorectal cancer cell proliferation, migration and invasion by targeting OTUB1. Am J Cancer Res 2017, 7, 159-172.
Is the data reproducible in other cell lines or in vivo?
Response: We are conducting translational research to support the proof of concept that podocyte-derived EVs lead to tubular cell apoptosis.
Patients with FSGS and MCD were enrolled and their urinary EV miRNA profiles are being evaluated. We hope to publish this data soon.
Round 2
Reviewer 2 Report
No further comments.